# Fatigue in Inflammatory Joint Diseases

**DOI:** 10.3390/ijms241512040

**Published:** 2023-07-27

**Authors:** Grzegorz Chmielewski, Michał S. Majewski, Jakub Kuna, Mateusz Mikiewicz, Magdalena Krajewska-Włodarczyk

**Affiliations:** 1Department of Rheumatology, School of Medicine, Collegium Medicum, University of Warmia and Mazury in Olsztyn, 10-900 Olsztyn, Poland; gchmielewski.gc@gmail.com (G.C.); kuna.jakub@wp.pl (J.K.); 2Department of Pharmacology and Toxicology, Faculty of Medicine, University of Warmia and Mazury in Olsztyn, 10-082 Olsztyn, Poland; michal.majewski@uwm.edu.pl; 3Department of Pathological Anatomy, Faculty of Veterinary Medicine, University of Warmia and Mazury in Olsztyn, 10-719 Olsztyn, Poland; mateusz.mikiewicz@uwm.edu.pl

**Keywords:** fatigue, inflammation, cytokines, inflammatory joint diseases

## Abstract

Fatigue is a prevalent symptom in various rheumatic diseases, such as rheumatoid arthritis, psoriatic arthritis, and ankylosing spondylitis. It is characterised as a subjective, enduring feeling of generalised tiredness or exhaustion, impacting the patient’s life quality and exacerbating disability. The fatigue nature is multifaceted, encompassing physiological, psychological, and social factors, and although the exact cause of inflammatory joint diseases is not fully understood, several factors are believed to contribute to its development. Despite high prevalence and importance, the symptom is often underestimated in clinical practice. Chronic inflammation, commonly associated with rheumatic diseases, has been proposed as a potential contributor to fatigue development. While current treatments effectively target inflammation and reduce disease activity, fatigue remains a persistent problem. Clinical evaluation of rheumatic diseases primarily relies on objective criteria, whereas fatigue, being a subjective symptom, is solely experienced and reported by the patient. Managing fatigue in inflammatory joint diseases involves a multifaceted approach. Identifying and comprehensively assessing the subjective components of fatigue in individual patients is crucial for effectively managing this symptom in everyday clinical practice.

## 1. Introduction

Fatigue is defined as an overwhelming, exhausting, and persistent feeling of tiredness that reduces the ability to function and perform daily activities. Fatigue occurs in many chronic conditions and diseases, such as cancer, inflammatory bowel disease, inflammatory joint disease, connective tissue diseases, autoimmune diseases, chronic obstructive pulmonary disease, chronic heart failure, and chronic kidney disease [1]. In some of these diseases, the cause of fatigue can be attributed to changes in muscle metabolism or cardiovascular system disorders, but in some conditions, such as multiple sclerosis or chronic fatigue syndrome, determining the pathomechanism is difficult. Numerous studies have demonstrated the impact of inflammatory cytokines on the central nervous system (CNS), but the exact mechanism by which the inflammatory state affects fatigue remains unclear [2]. Many factors unrelated to the inflammatory state have an impact on the development of fatigue, including hydration status, pain, concomitant therapy, physical activity, hypothyroidism, radiotherapy, lung capacity, blood pressure, heart rate, and left ventricular ejection fraction [3]. Research on fatigue is challenging because it has many clinical aspects and a very broad definition. Fatigue can be divided into physiological and pathological types. Physiological fatigue occurs after significant physical exertion and serves as a signal for rest and recovery. Pathological fatigue does not decrease during rest [4]. Most research on fatigue and the mechanism of its development is conducted in patients with cancer. Currently, the pathomechanism of fatigue is thought to be related to inflammation, disturbances in the hypothalamic–pituitary–adrenal axis, and activation of the autonomic nervous system [1]. Fatigue can be peripheral or central. Central fatigue can develop as a result of CNS disorders, prolonged physical exertion, or disturbances in neurotransmitter transmission in the CNS. Peripheral fatigue results from disturbances in the neuromuscular axis that transmit signals to the actin–myosin complex responsible for muscle contraction. In rheumatologic diseases, fatigue may be the result of both peripheral and central fatigue mechanisms [5]. Fatigue is often associated with pain, and this association has been documented in many studies. High levels of pain are often associated with significant fatigue [4]. In patients with inflammatory joint diseases, where pain is the main symptom, attempts have been made to link fatigue with disease activity. However, these associations are unclear and require further research. Numerous studies have shown that disease activity alone accounts for only a small portion of perceived fatigue, and medications that reduce disease activity do not always contribute to reducing fatigue [6]. The studies examining fatigue, disease activity, and inflammatory markers show conflicting results. Nevertheless, we concur that fatigue might be linked to combined measures of disease activity and is not consistently predicted by conventional inflammation markers like C-reactive protein (CRP) or erythrocyte sedimentation rate (ESR), which has been supported by other research [7,8,9]. For a long time, fatigue was not recognised as a significant clinical factor, and the impact of fatigue on the patient’s quality of life was not taken into account. During a 2002 Outcome Measures in Rheumatology meeting (OMERACT), the significance of fatigue as a symptom of rheumatic diseases and its impact on patients’ quality of life was officially acknowledged. This recognition emphasised the importance of considering fatigue when evaluating treatment effectiveness and assessing the overall disease burden [10]. Since then, fatigue has been considered an important clinical symptom in many diseases and a significant therapeutic target. Outcome Measures in Rheumatology is an international scientific initiative aimed at establishing standards for evaluating the effectiveness of treatments for rheumatic diseases [11]. In inflammatory joint diseases, fibromyalgia is a commonly coexisting condition. Fibromyalgia is a chronic and complex disorder characterised by widespread musculoskeletal pain, tenderness, and increased sensitivity to touch. Other symptoms include fatigue, mood changes, and sleep disturbances [12]. In the general population, the prevalence of fibromyalgia is estimated to be around 1 to 6% [13]. In inflammatory joint diseases, the prevalence of fibromyalgia is significantly higher than in the general population, and it oscillates to approximately 18–24% in rheumatoid arthritis (RA), 14–16% in Axial Spondyloarthritis (axSpA), and 18% in psoriatic arthritis (PsA) [14]. The coexistence of fibromyalgia and joint diseases poses diagnostic and therapeutic challenges. Additionally it also decreases the quality of life and provide difficulties in performing daily activities. Due to the frequent coexistence of fatigue with mood disorders, antidepressant medications such as tricyclic antidepressants or selective serotonin reuptake inhibitors (SSRIs) and serotonin–norepinephrine reuptake inhibitors (SNRIs) can be helpful in therapy [15]. Meta-analyses have shown that antidepressant drugs lead to a reduction in the concentration of pro-inflammatory cytokines in the blood, such as interleukin-1beta (IL-1beta), interleukin-4 (IL-4), and interleukin-6 (IL-6), which can result in fatigue reduction [16]. Selective serotonin reuptake inhibitors exhibit the most beneficial effects in restraining the inflammation markers in depression [17]. Sleep disorders can have a significant impact on fatigue, and addressing them can result in improvements in overall well-being and energy levels. Studies have shown that patients with RA who have lower sleep quality tend to experience higher levels of fatigue [18]. The use of medications which improve the quality and duration of sleep can significantly contribute to reducing fatigue, but there is a lack of in-depth research on this topic.

## 2. Fatigue and Inflammation

In chronic inflammatory diseases, fatigue develops as a result of the interaction between the immune system and the nervous system through pro-inflammatory cytokines. Cytokines are small proteins produced by lymphocytes, macrophages, microglial cells, astrocytes, and neurons involved in signalling pathways through which cells communicate with each other via autocrine, paracrine, and endocrine routes [3]. In inflammatory joint diseases, fibromyalgia is a commonly coexisting condition. Fibromyalgia is a chronic and complex disorder characterised by widespread musculoskeletal pain, tenderness, and increased sensitivity to touch. Other symptoms include fatigue, mood changes, and sleep disturbances [12,19]. When an infectious agent is present in the body, the cells of the immune system respond by producing pro-inflammatory cytokines. These cytokines can interact with the brain and trigger the development of sick behaviour [20]. It is widely believed that pro-inflammatory cytokines such as TNF-alpha, IL-1beta, and IL-6 are responsible for triggering sickness behaviour, which includes symptoms such as malaise, fever, loss of appetite, decreased libido, depressed mood, lethargy, pain hypersensitivity, excessive sleepiness, fatigue, and anhedonia [21]. The adaptive behavioural changes that develop in sick individuals are designed to prioritise the body’s fight against pathogens. During an infection, the immune response requires a significant amount of energy, making it energetically costly for the body. As a result, the body prioritises the allocation of energy and limits the expenditure on less essential behaviours during this time. Cytokines coordinate the immune response against the pathogen but also impact the CNS, leading to behavioural changes, including fatigue [2]. The structure separating the nervous system from the bloodstream is the blood–brain barrier (BBB). It is a semi-permeable barrier composed of endothelial cells of capillaries, astrocytes, and pericytes. The endothelial cells that compose this barrier are interconnected in a distinct manner compared to capillaries in other organs. They form tight junctions, which create a tight seal between the cells, ensuring a highly selective barrier function. These tight junctions create a close connection between cells, allowing the transport of substances from the plasma only through intracellular transport mediated by specific transport proteins or by diffusion (limited to hydrophobic molecules such as O_2_, CO_2_, hormones, or small nonpolar molecules). Microglial cells are one of the key cells ensuring the integrity of the BBB. Microglial cells located adjacent to the endothelial cells produce the protein claudin-5, which forms tight junctions [22]. Inflammation occurring in the periphery can be transferred to the CNS through several pathways. Firstly, pro-inflammatory cytokines can activate immunocompetent microglial cells located along the brain’s blood vessels. In a prolonged inflammatory state, microglial cells start the phagocytosis of the end feet of astrocytes, which are part of the BBB, leading to its permeability [23]. Additionally, in circumventricular organs, capillaries have a fenestrated structure, making the BBB permeable. This allows substances like interleukin-1alpha (IL-1alpha) to pass through the barrier via diffusion [24]. For certain cytokines such as IL-1alpha, IL-1beta, and IL-1RA, there is a saturable transport system that allows these molecules to penetrate from the circulation into the CNS. Additionally, animal studies have shown that TNF-alpha can induce changes in the endothelial cells forming the BBB. Under the influence of this cytokine, the expression of tight junction proteins is reduced, resulting in increased intercellular distances and contributing to increased permeability of the BBB [25]. The last possible way to transfer inflammation to the CNS is through the production of cytokines by locally activated perivascular endothelial cells and macrophages in the brain [26] (Figure 1).

Local cytokine production spreads in the brain through diffusion and activation of subsequent microglial cells, as well as via projecting neurons. Additionally, apart from activation through the humoral pathway, there is also a neural pathway for transferring inflammatory states from the periphery to the CNS. Pro-inflammatory cytokines interact with afferent nerve fibres and the autonomic nervous system, such as the vagus nerve, glossopharyngeal nerve, afferent fibres in the gastrointestinal tract (the gut–brain axis), and joints [2]. It appears that the activation of microglial cells and their production of pro-inflammatory cytokines in the CNS require the interaction of two mechanisms: a faster pathway through afferent nerve fibres and a slower cytokine pathway. All of the pathways through which communication occurs between the immune and nervous systems converge in a single anatomical structure known as the dorsal vagal complex [24]. The afferent fibres of the vagus nerve mediate the transmission of pro-inflammatory signals from internal organs such as the peritoneum, lungs, and intestines to the nucleus tractus solitarius (NTS). The NTS has connections with multiple brain regions, and the transferred pro-inflammatory signal leads to increased expression of pro-inflammatory cytokines in brain areas that play a significant role in the regulation of sleep and fatigue processes [3]. Activation through afferent fibres also sensitises target structures in the brain to cytokines produced in circumventricular organs [20]. When cytokines enter the CNS through the described pathways, they can induce changes in the neuroendocrine system and neurotransmitters, resulting in disruptions in brain function and alterations in behaviour [2]. As a result of the pro-inflammatory cytokines entering the CNS, microglial cells and astrocytes become activated. This leads to the development of inflammation in the nervous system, known as neuroinflammation, and subsequently triggers local cytokine production and the activation of signalling pathways such as nuclear factor (NF)-kappaB, Janus kinase (JAK)-signal transducer and activator of transcription (STAT), and mitogen-activated protein kinase (MAPK) pathways [27]. This sequence of events forms the basis for the development of fatigue and its symptoms as a result of immune system activation. The rapid progress in neuroimaging techniques has provided scientists with the ability to precisely identify brain structures that play a role in the sensation of fatigue. This advancement allows for a more accurate understanding of the neural mechanisms underlying fatigue and its development. Depending on the type of fatigue experienced, the involvement of specific brain structures may also vary. When considering the regions of the CNS involved in the development of cognitive fatigue, the following areas are noteworthy: the striatum of the basal ganglia, the ventro-medial prefrontal cortex (vmPFC), the dorsal anterior cingulate cortex (dACC), the dorsolateral prefrontal cortex (DLPFC), and the anterior insula [28]. The subcortical structures, specifically the basal ganglia and the frontal cortex, form a dopaminergic network known as the frontostriatal dopaminergic neurocircuitry. This network plays a significant role in mediating behavioural changes in response to inflammatory cytokines, particularly in the context of reward-based decision-making [29]. Disruptions occurring in the frontostriatal reward circuitry, which includes the striatum, thalamus, vmPFC, and DLPFC, play a crucial role in the development of fatigue [30]. Indeed, other researchers focus on disruptions in the basal ganglia. The basal ganglia, known for their essential role in regulating motor activity and motivation, are considered a pivotal brain region associated with fatigue development. Fatigue frequently presents in conditions characterised by alterations or damage to the basal ganglia, such as Parkinson’s disease, multiple sclerosis, or human immunodeficiency virus (HIV) infection. Impairments to the basal ganglia can result in fatigue and psychomotor slowing. These disruptions can adversely affect the functioning of the motor system and contribute to the fatigue experienced in individuals affected by these conditions [31]. Research conducted on healthy volunteers experiencing fatigue has revealed in which brain regions the changed activity occurred. Moreover, the relationship between fatigue level, motivation, and increased mean diffusivity (MD) in the right hemisphere’s putamen, globus pallidus, and the caudate nucleus has been defined. These regions, which are part of the basal ganglia, play a role in motor control and motivation [32]. Proinflammatory cytokines in the basal ganglia can lead to changes in the concentrations of neurotransmitters such as noradrenaline, dopamine, serotonin, melatonin, and glutamate, which, in turn, result in alterations in behaviour. The neurotransmitter dysregulation can impact various aspects of behaviour, including mood, motivation, cognition, and sleep–wake cycles. The interplay between cytokines and neurotransmitters in the basal ganglia contributes to the complex mechanisms underlying behavioural changes associated with inflammatory processes [27]. Pro-inflammatory cytokines exert their most significant mechanism of action on neurotransmitter production by influencing the enzymes responsible for synthesising serotonin and dopamine. Pro-inflammatory cytokines have the ability to enhance the enzymatic activity of indoleamine 2,3-dioxygenase (IDO) and guanosine triphosphate–cyclohydrolase-1 (GTP-CH1) while reducing the effectiveness of the cofactor tetrahydrobiopterin (BH4). These actions ultimately lead to a decrease in the levels of serotonin and dopamine. Consequently, the altered neurotransmitter levels contribute to the development of sick behaviour [33]. Among pro-inflammatory cytokines, it has been shown that interferon-alpha (IFN-alpha) strongly induces the gene encoding the enzyme IDO. The enzyme indoleamine 2,3-dioxygenase plays a role in the breakdown of tryptophan. It converts tryptophan into kynurenine, which can further be metabolised into neuroactive compounds such as quinolinic acid. Quinolinic acid, when present in excessive amounts, can have toxic effects on the CNS [34]. Studies conducted on fatigue patients with cancer have demonstrated that heightened activity of IDO results in reduced levels of tryptophan and an elevated ratio of kynurenine to tryptophan (Kyn/Trp). This shift in metabolites has been associated with increased fatigue [35]. The decrease in tryptophan level leads to a reduction in serotonin production. A deficiency of serotonin can contribute to various symptoms, including fatigue, increased sensitivity to pain, aggression, and depressive disorders [36]. Another important enzyme influenced by pro-inflammatory cytokines is GTP-CH1. Activation of this enzyme leads to the synthesis of neopterin and depletion of the substrate for the production of the important cofactor BH4, which is essential for dopamine and serotonin synthesis [2]. It might seem that the use of drugs which increase dopamine release or block dopamine reuptake would be helpful in decreasing fatigue. However, clinical studies have shown limited effectiveness of these medications in treating fatigue [37].

### 2.1. Interleukin-1

The interleukin-1 family consists of eleven cytokines that play an important role in regulating immune and inflammatory responses. The first and best-studied cytokines in this family are IL-1alpha and IL-1beta, which induce similar pro-inflammatory reactions by interacting with the interleukin-1 type 1 receptor (IL-1R1) [38]. IL-1alpha is mainly expressed on the surface of monocytes and B lymphocytes, while IL-1beta is a soluble cytokine secreted by various cells, including macrophages, monocytes, and dendritic cells [39]. IL-1beta can also bind to the interleukin-1 type 2 receptor (IL-1R2), which acts as a decoy receptor, inhibiting the activity of IL-1beta [3]. Another mechanism that regulates the activity of IL-1 is the production of interleukin-1 receptor antagonists (IL-1RA—a competitive inhibitor) [40]. Many studies in animal models have demonstrated that IL-1beta plays a crucial role in regulating the development of fatigue. Administering IL-1beta systemically or directly into the brain resulted in the manifestation of sick behaviour [41]. Sickness behaviour is characterised by sleepiness, loss of appetite, decreased activity, and social withdrawal. Behavioural changes are theorised to increase the chances of survival during infection [39]. Sickness behaviour is analogous to fatigue in humans and is manifested by disruptions in physical activity, appetite, and sleep [5]. Additional evidence supporting the influence of IL-1 on the development of sickness behaviour comes from studies on mice lacking IL-1 receptors. In these animals, the sickness behaviour development did not occur, regardless of whether IL-1beta was administered intraperitoneally or directly into the brain [42]. Studies on animals have confirmed the influence of IL-1beta on motivational symptoms such as psychomotor slowing, anergia, and fatigue. However, administering IL-1beta to rats in doses that did not induce a general sickness state led to a decrease in their willingness to work for food [43]. In another study using an animal model of RA induced by collagen, an increase in the number of microglial cells in the area postrema and elevated levels of IL-1beta were observed. The area postrema is a region in the brain where the BBB is least restrictive, allowing inflammatory signals from the periphery to enter the CNS. This study suggests that chronic inflammatory conditions like RA may impact brain regions with greater permeability and lead to neurological consequences [44]. In humans, intravenous administration of IL-1beta has been shown to lead to the development of chills, headaches, fatigue, and hypotension [45]. Studies involving the IL-1 receptor antagonist anakinra provide compelling evidence for the influence of IL-1 on the development of fatigue. These studies have consistently shown that administering anakinra to patients with RA not only reduces disease activity but also leads to a significant and enduring reduction in fatigue levels [46]. Regrettably, studies attempting to measure the concentration of IL-1beta often fail to yield the desired results. Even in cases of high inflammation, the blood concentration of this cytokine remains low. This is partly due to the fact that a large portion of IL-1beta is accumulated within cells, and the circulating molecules in the bloodstream are bound to the IL-1R2 [26]. The concentration of IL-1beta is low under normal conditions of homeostasis in the nervous system. However, this state changes when the body experiences stress, leading to an increase in IL-1beta concentration. This elevated concentration of IL-1beta starts to exert detrimental effects on the body [47]. IL-1 influences the CNS through various pathways, and one of them is associated with afferent fibres of the vagus nerve. IL-1beta activates these fibres, which terminate in the NTS. Studies have shown that intraperitoneal administration of IL-1beta and lipopolysaccharide (LPS) leads to the development of fever, anorexia, increased levels of corticosteroids in the bloodstream, and alterations in norepinephrine levels in the brain. These effects occur through the pathway involving the vagus nerve [48]. On the other hand, in animals that have undergone vagotomy, fever has been absent even after the administration of low doses of IL-1beta or endotoxin [49]. The development of fever after the administration of high doses of IL-1beta and endotoxin can be attributed to several reasons. According to researchers, the transmission of inflammation through the vagus nerve is crucial in the early stages of infection when the concentration of pro-inflammatory cytokines is elevated only locally. Later, when cytokine levels in the bloodstream become systemic, the bloodstream pathways may play a dominant role in transmitting the inflammatory state to the CNS [48]. The results of these various studies suggest that there is no single dominant pathway through which the immune system interacts with the nervous system. Instead, complex interactions and communication network between the two systems have been discovered. Different pathways may be involved depending on the specific context and conditions of the immune response. The interplay between the immune and nervous systems is a dynamic and multifaceted process that is still being actively researched and understood [24].

### 2.2. Interleukin-6

IL-6 is a pro-inflammatory cytokine that has diverse actions. It is produced in response to infection or tissue damage and contributes to the body’s defence mechanisms by stimulating the acute phase response, immune reactions, and haematopoiesis [50]. IL-6 plays a role in the differentiation and maturation of B lymphocytes and in the differentiation of CD4+ T cells into Th17 cells. The protein is primarily produced by macrophages under the influence of IL-1 and TNF-alpha. Interestingly, IL-6 itself can cause a decrease in the production of these cytokines [39]. Furthermore, IL-6 plays an auxiliary role in generating fever through IL-1beta and enhances the production of CRP in the liver [51]. Additionally, this cytokine is involved in the neuroendocrine system, insulin resistance, lipid metabolism, neuropsychological behaviours, and communication between the immune system and the CNS [52]. In many studies involving cancer patients, elevated levels of IL-6 have been associated with higher levels of fatigue [53]. Administration of IL-6 to healthy individuals resulted in sleep disturbances, insomnia, and fatigue [54]. The impact of IL-6 on fatigue development seems to be supported by studies using drugs that block this cytokine. In the OPTION study, it was demonstrated that patients receiving tocilizumab experienced a significant reduction in fatigue compared to the placebo group [55]. Similar results were obtained in the TOWARD study [56]. Another IL-6 inhibitor, sarilumab, has also shown efficacy in reducing fatigue in the MOBILITY study [57]. IL-6 plays an important role in the pathomechanism of fatigue not only in inflammatory joint diseases but also in other conditions. It has been shown that IL-6 levels are significantly higher in patients with Parkinson’s disease and fatigue compared to patients without fatigue [58].

### 2.3. Other Cytokines

A particularly pronounced impact of pro-inflammatory cytokines on the functioning of the CNS, as well as their influence on behaviour and mood, has been observed in patients treated with IFN-alpha for cancer or viral hepatitis C. IFN-alpha, as a proinflammatory cytokine, activates the immune system and is associated with numerous neuropsychiatric complications. During the hepatitis C treatment with IFN-alpha, nearly 80% of patients experienced a rapid onset of fatigue. Other symptoms, such as depressive mood and cognitive impairments, occurred less frequently and were observed after prolonged therapy. This suggests a stronger biological association between fatigue and inflammation [2]. In their study, Felger et al. demonstrated that IFN-alpha therapy leads to a reduction in the conversion of phenylalanine to tyrosine, resulting in decreased dopamine levels in the cerebrospinal fluid and higher levels of fatigue in patients [59].

## 3. Fatigue in Rheumatoid Arthritis

Fatigue is present in approximately 40–70% of patients with RA [6]. Statistically, significantly more women than men with RA experience fatigue, which may be explained through neurophysiological differences [60]. Fatigue is often the main issue for patients with RA, and it has a significant impact on their life quality [61]. The physical symptoms of RA, such as pain, inflammation, and fatigue, have an impact on mental health [62]. There are many scales for disease activity and physical function assessment recommended to use in patients with RA. However, consistent criteria for diagnosing and treating fatigue in RA patients have not been established. Various scales are used to determine the level of fatigue, ranging from the simplest scale like the 0–100 mm visual analogue scale (VAS) to more advanced tools such as the Functional Assessment of Chronic Illness-Fatigue (FACIT-F) questionnaire, the Bristol Rheumatoid Arthritis Fatigue-Multidimensional Questionnaire (BRAF-MDQ), and the Patient Reported Outcome Measurement Information System (PROMIS)-29 Fatigue T-score [63]. In recent years, three scales have been approved for assessing fatigue in patients with rheumatoid arthritis: the Bristol Rheumatoid Arthritis Fatigue Multidimensional Questionnaire (BRAF-MDQ), the revised Bristol Rheumatoid Arthritis Numerical Rating Scales (BRAF-NRS V2), and the Rheumatoid Arthritis Impact of Disease (RAID) scale. These scales take into account various aspects of fatigue and the consequences of disease symptoms [64] (Table 1).

Unfortunately, in the past, fatigue was only considered as an endpoint in a few studies. For many years, fatigue occurring in RA was regarded as a consequence of the disease process, a complication of pain and joint symptoms, or as an adverse effect of the treatment being administered [6]. It is currently believed that fatigue in RA arises from the interplay of multiple factors, which can be grouped into three categories: disease-related factors (inflammation, pain, sleep disturbances, disability), individual factors (comorbidities), and cognitive–behavioural factors (personality, activity levels) [63]. The association between inflammatory markers and disease activity with fatigue is unclear and inconsistent. In some studies, strong correlations have been found between inflammation and fatigue, while in others, no influence on fatigue levels has been observed [65]. Fatigue can also occur in patients with low disease activity. This may be due to the chronic pain experienced by patients with RA, which can lead to sleep disturbances and mood disorders such as depression [63]. Indeed, these factors appear to be more strongly associated with fatigue than disease activity alone. The presence of chronic pain, sleep disturbances, and mood disorders can have a significant impact on the experience of fatigue in patients with RA. Addressing these factors, along with managing disease activity, can be crucial in effectively managing fatigue in RA patients [66]. Other researchers have shown that fatigue is positively correlated with pain, CRP, Disease Activity Score-28 (DAS-28), and ESR but not with the ratio of swollen to tender joints, the ratio of tender to swollen joints, or disease duration [67]. The pain appears to be the factor with the greatest influence on fatigue, particularly the pain hypersensitivity present in some patients with RA. The use of analgesic medications should be as important as disease-modifying therapy [68]. After examining 192 patients with RA and assessing the fatigue level using the FACIT-F questionnaire, Wagan et al. obtained results indicating that fatigue was associated with age, education level, concomitant hypertension, hepatitis C virus (HCV) infection, and the DAS-28. An increase of one unit on the DAS-28 scale resulted in a 2.71 unit increase in the level of fatigue [69]. The reduction in fatigue is more likely attributed to improved pain management and alleviation of depressive symptoms rather than a direct decrease in inflammatory activity [65]. The studies comparing fatigue in seropositive and seronegative RA are limited. No differences in fatigue levels were observed between patients with seropositive and seronegative RA [70,71]. In patients with RA, IL-1beta plays a significant role in the fatigue pathogenesis. Numerous studies have shown its association with fatigue. Lampa et al. demonstrated significantly increased levels of IL-1beta and decreased levels of the anti-inflammatory IL-1RA in the cerebrospinal fluid (CSF) of RA patients compared to a healthy control group. Additionally, it was found that the concentration of IL-1beta in the CSF was higher than in the serum. The concentration of TNF-alpha in the CSF of RA patients was not elevated compared to healthy individuals, while the concentration of IL-6 was slightly increased. Furthermore, this study showed that the increased concentration of IL-1beta in the CSF was associated with higher fatigue levels and sleep problems. This may support the hypothesis that in RA, in addition to joint inflammation or systemic inflammation, there is also an immunological activation in the CNS [72]. The use of biological drugs not only contributes to better control of RA, but several studies also confirm their impact on reducing fatigue. The use of anti-TNF-alfa drugs and other biological agents leads to a reduction in fatigue to a mild to moderate degree. However, it remains unclear whether this is a direct effect of biological drugs on fatigue or an indirect effect through their impact on inflammatory status or disease activity [73]. After analysing a registry of patients with RA, Druce et al. found a significant reduction in fatigue among patients treated with TNF-alpha inhibitors. Additionally, the study identified initial factors that may influence the reduction of fatigue with anti-TNF therapy. According to the study, female gender, occupational activity, low level of disability, seropositivity, non-use of glucocorticosteroids, absence of hypertension, and good mental health are characteristics that have an impact on reducing fatigue with the use of TNF-alpha inhibitors [74]. In other studies, the significant impact of TNF-alpha inhibitor therapy on fatigue has not been consistently confirmed. It appears that beyond their anti-inflammatory effects, TNF-alpha inhibiotrs do not have a substantial impact on persistent fatigue [75]. In a study using tocilizumab (an anti-IL-6 agent) in patients with RA, a significant reduction in fatigue perception was observed after 24 weeks. The reduction in fatigue was also correlated with a decrease in disease activity. Improvement in sleep and reduction in depressive symptoms have a greater impact on changes in fatigue levels [76]. Janus kinase inhibitors are the newest drugs used in RA and other inflammatory joint diseases. Janus kinases (JAK1, JAK2, JAK3, and tyrosine kinase 2/TYK2) are enzymes that participate in transmitting information from cytokine and growth factor receptors (e.g., TNF-alpha, IL-1beta, IL-6, IL-12/IL-23, interleukin-17 (IL-17), and IFN-gamma) located on the cell membrane to the interior of the cell. Their function involves phosphorylating (activating) signal transducer and activator of transcription (STAT) proteins, which are responsible for further signal transmission to the cell nucleus and initiating the process of protein transcription. The JAK-STAT signalling pathway plays a significant role in haematopoiesis, the development of inflammation, and the functioning of the immune response. JAK inhibitors like tofacitinib (blocking JAK1, JAK3), baricitinib (blocking JAK1, JAK2), upadacitinib (blocking JAK1), and filgotinib (blocking JAK1) have shown significant advantages in reducing inflammation, pain, and fatigue, leading to improvements in life quality [77]. However, further studies are needed to conclude which of the mentioned drugs will most effectively reduce fatigue.

## 4. Fatigue in Ankylosing Spondylitis

In ankylosing spondylitis (AS), fatigue, along with pain and morning stiffness, is the most commonly experienced symptom of the disease. It occurs in approximately 50–70% of patients, and it is strongly associated with pain and can be difficult to alleviate or cure [78]. The combination of these two conditions results in limited mobility and functioning for the patient, leading to reduced life quality and work productivity. The exact fatigue cause in AS is not fully understood, but several factors may contribute to its development. The chronic inflammation associated with AS, as well as the pain and stiffness in the spine and joints, can contribute to increased fatigue levels. Additionally, the immune system activation and the presence of depression or sleep disturbances can further exacerbate fatigue in AS [79]. The level of fatigue is one of the components used to assess disease activity in the Bath Ankylosing Spondylitis Disease Activity Index (BASDAI), which is one of the most commonly used scales in patients with AS. This highlights the clinical significance of this symptom. Another reliable method to assess fatigue in AS is the use of the FACIT-F questionnaire [80]. Numerous associations have been demonstrated between disease activity and fatigue. Patients with higher fatigue levels tend to have higher disease activity, elevated inflammatory markers, and reduced life quality [81]. Connolly et al. have demonstrated that higher fatigue levels were associated with reduced work productivity, increased disease activity, pain, and decreased physical functioning [82]. Higher values of the waist-to-hip ratio, BASDAI, and sleep disturbances are independent predictive factors for the occurrence of fatigue in patients with AS [83]. Elevated levels of pro-inflammatory cytokines such as TNF-alpha, IL-17, and IL-23 in patients with AS may indicate their influence on the development of inflammatory pain, fatigue, and depression. Fatigue is strongly correlated with pain compared to other factors such as age, depression, anxiety, physical activity, and sleep disturbances [84]. Specific regions of the brain where structural and functional alterations were detected in patients with AS and fatigue. These regions include the sensory/somatomotor network (SMN), dorsal attention network (DAN), and task control network (TCN). Changes in brain activity and connectivity within these networks may contribute to the experience of fatigue in individuals with AS [85]. Treatment with TNF-alpha inhibitors in patients with AS effectively reduces disease activity and alleviates fatigue. However, fatigue reduction was not optimal for the majority of patients, and approximately 80% of them still experienced fatigue. While TNF-alpha inhibitors can be effective in managing AS, it is important to note that fatigue may persist in some individuals despite treatment [81]. Anti-TNF-alpha therapy results in a significant but minimal reduction in fatigue in axSpA. The greatest improvement is observed in patients with sleep disturbances [86]. Better results in combating fatigue have been obtained in patients treated with IL-17 blockers. In the MEASURE 1 and 2 studies in which secukinumab was used, a reduction in fatigue assessed by the FACIT-F questionnaire was observed compared to the placebo group. The fatigue reduction was correlated with clinical improvement assessed by disease activity scales such as Assessment of Spondyloarthritis International Society (ASAS) 20/40, ASAS5/6, Ankylosing Spondylitis Disease Activity Score (ASDAS)-CRP, BASDAI, and Short-Form 36. The fatigue reduction was noticeable as early as four weeks after initiating secukinumab treatment and was sustained throughout the therapy period in 75–81% of patients [87]. The use of secukinumab leads to a rapid and sustained reduction in pain and fatigue, regardless of the initial level of CRP and previous TNF-alpha inhibitor therapy [88]. Another IL-17 blocker, ixekizumab, also contributes to the reduction of fatigue compared to placebo in both biologic-naïve patients and those who were previously ineffective in TNF-alpha inhibitor treatment [89]. Janus kinase inhibitors are another group of drugs used in AS that have shown effectiveness in fatigue reduction. In a study by Navarro-Compán et al., patients receiving tofacitinib experienced a significant fatigue reduction measured by the FACIT-F questionnaire and the BASDAI scale after 2 weeks of therapy [90]. The use of upadacitinib is associated with a significant fatigue reduction; however, this effect requires long-term therapy and can be observed after approximately 14 weeks [91].

## 5. Fatigue in Psoriatic Arthritis

Fatigue is a common problem in patients with PsA. It often coexists with sleep disturbances, depressive disorders, and anxiety [92]. Fatigue is one of the most commonly reported symptoms in PsA, along with joint pain and skin changes, which significantly impact life quality. Fatigue greatly affects daily activities, physical functioning, work productivity, and social interactions [93]. Moderate to severe fatigue is experienced by approximately 30–50% of patients with PsA [94]. Fatigue is more common in patients with PsA compared to those with psoriasis only [95]. Recently, researchers have started to pay more attention to fatigue as a crucial aspect of PsA. In 2016, the OMERACT expert group recommended the inclusion of fatigue as one of the domains of PsA in clinical and observational studies. Various scales and questionnaires are used to assess fatigue in psoriatic arthritis, including the VAS, numerical rating scale, Medical Outcomes Study Short Form 36-item (SF-36), Multidimensional Assessment of Fatigue (MAF) scale, FACIT-F, and Fatigue Severity Scale (FSS) [96]. To assess the fatigue level in PsA, the BRAF-NRS scale can also be used, which is commonly used in RA [97]. Many studies have shown that disease activity is the strongest fatigue predictor in PsA, and fatigue is lowest in patients in remission [98]. High fatigue levels are associated with a greater number of affected joints and high pain levels [99]. The association between fatigue and disease activity has been confirmed in various scales used to assess disease activity. Tin Lok Lai et al. described that the occurrence and intensity of fatigue are related to disease activity measured by the Disease Activity in Psoriatic Arthritis (DAPSA) scale and Psoriasis Area and Severity Index (PASI) scale, while it was not correlated with age, gender, disease duration, treatment, or comorbidities. In another study, it was shown that fatigue severity was associated with poorer health status, physical functioning, work productivity, and health-related life quality. Patients reporting higher levels of fatigue were older, had higher disease activity and perceived pain levels, had a longer time since diagnosis and symptom duration, and had more frequent anxiety/depressive symptoms [100]. Reduction in disease activity is associated with lower fatigue levels, and decreased fatigue levels were observed in 81.5% of patients with very low disease activity (VLDA). Fatigue in PsA is linked to several factors, including reduced educational achievement, heightened severity of inflammatory joint disease resulting in failure to attain minimal disease activity (MDA), presence of comorbidities, joint deformities, and physical disability [97]. Independent fatigue predictors include disease activity assessed by DAPSA, IL-17 levels, and reduced life quality measured by the Psoriatic Arthritis Quality of Life (PsAQOL) questionnaire [101]. Analysing a registry of 880 Danish patients with PsA confirmed that fatigue was associated with higher baseline disease activity, pain, lower life quality, and comorbidities. Furthermore, it has been shown that TNF-alpha inhibitors treatment leads to a fatigue reduction, but the improvement is not satisfactory, and fatigue remains a dominant symptom, contributing to an increased risk of discontinuing anti-TNF-alpha therapy [102]. Indeed, other studies have also confirmed the inadequate reduction of fatigue during TNF-alpha inhibitor treatment, which affects overall life quality, physical functioning, and work productivity. Despite the benefits of TNF-alpha inhibitors in managing disease activity, fatigue persistence as a prominent symptom highlights the need for additional interventions to specifically target and alleviate fatigue in these patients [103]. Meta-analysis studies have shown that biological disease-modifying antirheumatic drugs such as adalimumab, certolizumab pegol, sekukinumab, ustekinumab, and apremilast have a significant but minimal effect on fatigue in patients with PsA. These medications have been found to have a greater impact on reducing pain, suggesting that the relationship between disease activity and fatigue may be weaker compared to the relationship between disease activity and pain [104]. It appears that medications with different mechanisms of action may prove to be more effective in reducing fatigue. Administration of the IL-17 blocker, sekukinumab, has shown clinically significant improvement in disease activity and health-related life quality measures, as well as pain and fatigue reduction [105]. There are high hopes associated with the use of new targeted synthetic disease-modifying antirheumatic drugs (DMARDs)—Janus kinase inhibitors. Tofacitinib contributes to fatigue reduction in patients who have previously been treated with TNF-alpha inhibitors, as well as in the TNF-naive group [106]. A study utilising upadacitinib yielded comparable findings, demonstrating a reduction in fatigue [107]. In the latest studies, it has been observed that guselkumab, an IL-23 blocker, also provides clinically significant and sustained fatigue reduction [108]. The use of the latest drugs appears to be a promising prospect for patients experiencing fatigue in the course of PsA, but the issue requires further research.

## 6. Conclusions

Fatigue is a challenging symptom to diagnose, and its effective treatment can be a significant challenge. Managing fatigue requires a holistic approach that takes into account not only the effective treatment of inflammatory joint disease but also the appropriate support for the patient. The use of the latest drugs holds promise for improving the patients’ life quality. However, further research and treatment individualisation is necessary for patients with inflammatory joint disease and fatigue-related symptoms. Additionally, it is important to understand that fatigue in inflammatory joint disease is not solely the result of physical exertion or lack of sleep. It is a complex symptom that can have diverse causes, including inflammatory processes, hormonal imbalances, depression, anxiety, and the impact of the medication used in the therapy. Therefore, a multidisciplinary approach is necessary, which takes into account both the reduction of disease activity and the factors management related to mental health and life quality. Further research is crucial to better understand the mechanisms underlying fatigue and to explore new avenues for its management.

## Figures and Tables

**Figure 1 ijms-24-12040-f001:**
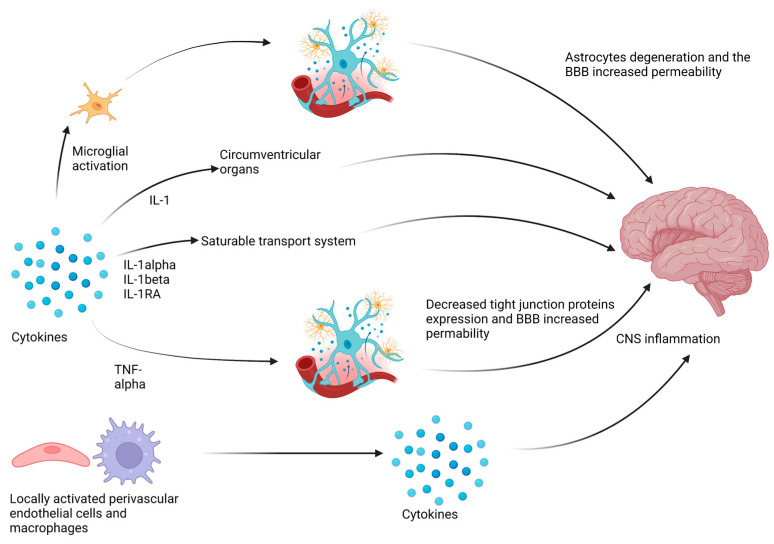
Inflammation transfer pathways to the central nervous system.

**Table 1 ijms-24-12040-t001:** Selected fatigue scales used in rheumatoid arthritis (RA).

Scale	Aspect Assessed	Number of Questions	Scale Type	Recall Period
Visual Analogue Scale (VAS)	Severity	1	Visual analogue	Usually past 7 days
Functional Assessment of Chronic Illness Therapy Fatigue Scale (FACIT-F)	Severity and impact	13	5-point Likert	Past 7 days
Bristol Rheumatoid Arthritis Fatigue-Multidimensional Questionnaire (BRAF-MDQ)	Severity and impact	20	4-point Likert, except for the first three items, which are numerical or categorical as appropriate	Past 7 days
revised Bristol Rheumatoid Arthritis Numerical Rating Scales (BRAF-NRS V2)	Severity and impact	3	0–10 point numerical	During past 7 days
Rheumatoid Arthritis Impact of Disease (RAID)	Severity and impact	7	0–10 point numerical	Past 7 days
Patient Reported Outcome Measurement Information System (PROMIS)-29 Fatigue T-score	Severity and impact	29	5-point Likert, except for the last question, which is 10-point Likert	Past 7 days

## Data Availability

No new data were created in this work.

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
