# Peer review of "Fatigue in Inflammatory Joint Diseases"

_ijms, 2023, doi:10.3390/ijms241512040_

Round 1

Reviewer 1 Report

The authors present an interesting review of the fatigue that often accompanies inflammatory rheumatic diseases. The manuscript is well written and treats the topic quite comprehensively.

However, some points need clarification:

1) Fatigue is a characteristic symptom of fibromyalgia. Fibromyalgia is a comorbidity that in many cases accompanies several inflammatory rheumatic diseases. The authors should clarify the presence of concomitant fibromyalgia in inflammatory rheumatic diseases and its role in the fatigue associated with these conditions.
2) It should be clarified whether fatigue is more frequent in serum-positive rheumatoid arthritis (APCA+) or seronegative rheumatoid arthritis.
3) The authors should better clarify possibly with their own commentary in what frequency fatigue is associated with the presence of elevated indices of inflammation (ERS and CRP) or is more frequently independent of these factors.
4) Line 487 contains an incorrect statement that should be corrected.
5) Since JAK inhibitors are mentioned, a brief description of the mechanism of action on the JAK/STAT transduction pathway is needed. Preferential inhibition of different cytokines among the various components of this drug class should also be specified.
6) The possible clinical effect of antidepressants (SSRI/SSNRI, tricyclic antidepressants, etc.) on fatigue should be commented on.
7) The treatment of sleep disorders with various drugs (antidepressants, pregabalin, benzodiazepines, Z-drugs etc.) should also be commented on to provide guidance for clinical practice in the treatment of fatigue associated with rheumatic diseases.

English needs only miror editing.

Reviewer 2 Report

In this review, the authors discussed detailed mechanisms of fatigue in inflammatory joint diseases. The review is interesting and well-written. It covers most aspects required in this topic.

The manuscript should be revised for typos (e.g. line 199), language mistakes (e.g. lines 414 - 415), and inconsistent use of abbreviations throughout the manuscript (e.g. lines 431, 464).

sekukinumab was mentioned as a TNF-alpha blocker (line 488) and as an IL-17 blocker as well (line 493). I suppose it is actually an IL_17 blocker.

Minor revision of the language is required.
